# “It’s Important but, on What Level?”: Healthy Cooking Meanings and Barriers to Healthy Eating among University Students

**DOI:** 10.3390/nu12082309

**Published:** 2020-07-31

**Authors:** Mercedes Vélez-Toral, Carmen Rodríguez-Reinado, Ana Ramallo-Espinosa, Montserrat Andrés-Villas

**Affiliations:** 1Department of Social, Developmental and Educational Psychology, University of Huelva, 21007 Huelva, Spain; maria.velez@dpee.uhu.es (M.V.-T.); ana.ramallo802@alu.uhu.es (A.R.-E.); montserrat.andres@dpsi.uhu.es (M.A.-V.); 2Department of Sociology, Social Work and Public Health, University of Huelva, 21007 Huelva, Spain

**Keywords:** healthy eating, healthy cooking, barriers, university students, focus group

## Abstract

The negative impact of a sedentary lifestyle and poor diet on health is evident across the lifespan, but particularly during the university period. Usually, the diet of university students is rich in sweetened drinks and processed foods and low in fruits, vegetables and legumes. Although there is an association between maintaining a healthy diet and the frequency of cooking at home, the time currently spent on cooking or learning how to cook is decreasing globally. The main aim of this study was to explore university students’ perceptions about healthy cooking and barriers to eating healthily. A group of 26 students participated in four focus groups. Content analysis was conducted using Atlas.ti v.8. Students perceived cooking healthily as a more complicated and time-consuming process than cooking in general. Individual and environmental factors were the most reported barriers. Costs and time, among others, were the main barriers pointed out by students with regard to healthy eating. This study highlights the need to develop interventions that modify these false perceptions about cooking healthily, and to train students so that they are able to cook healthy meals in a quick, easy, and cost-effective way. Further, specific actions are required in the university setting to minimize access to unhealthy options and to promote those linked to healthy eating.

## 1. Introduction

Linked to the evolution of lifestyles and food availability, eating habits and food patterns change constantly [1]. This has led to the increase in non-communicable diseases that represent a major public health problem in developed countries, including Spain [2]. It is estimated that in Spain, 80% of men and 55% of women will be overweight or obese by 2030 [3].

The negative impact of a sedentary lifestyle and a poor diet on health is present in all life stages [4]. However, in the university period there are particularly significant changes in lifestyle [5,6]. Beginning university and moving away from the family of origin to live independently involves a modification in eating habits [5,7]. Generally, the diet of these students is rich in sweetened drinks and processed foods and low in fruits, vegetables and legumes [7,8,9,10]. Therefore, it is not surprising that 20.6% of Spanish university students are overweight or obese [11].

In this particular context, in which the Mediterranean Diet is associated with good health and quality of life [12], we define healthy diet as the intake of a great amount of vegetable products, the use of olive oil as a main fat source, a frequent consumption of fresh fish, a moderate intake of dairy products, white meats and eggs, and a low consumption in frequency and quantity of red meats and processed meats [13]. Although there is an association between maintaining a healthy diet and the frequency of cooking at home [14,15,16], less time is currently being spent on cooking or learning how to cook in comparison with previous decades [17]. This is due to the economic and social changes that involved the incorporation of women into the labor market [17] and the fact that convenience foods are more readily accessible [18,19]. With regard to the latter, it is known that the greater availability of convenience or ultra-processed food means that cooking is not perceived as necessary to satisfy daily dietary needs [18]. Therefore, the frequent consumption of this type of food is related to the low consumption of food prepared at home, along with the decline of cooking skills and a decrease in the frequency with which such skills are used [20]. In Spain, recent years have seen a considerable decrease in the time spent cooking. For instance, in 2015, Spanish people dedicated an average of 8.66 h per week to cooking [21] versus 8.62 h per week in 2018 [22] and 10.25 a decade before [23]. Taking into account age, it is the group of 18–30 years that spends the least time cooking, investing 6.95 h per week [22], a figure below the national average. Additionally, the consumption of processed foods has increased, as shown by the data from a study conducted in Spain with university students in which 88.3% reported having consumed this type of food in between one and three occasions in the last three days [24].

Although cooking is a relevant contribution to diet quality [25], it is under-researched and not well understood [26], maybe due to the complexity of the term, which involves different abilities and procedures like cooking skills and food skills [27]. For this research, and in line with McGowan et al. [28], cooking skills are considered as a set of physical or mechanical skills used in food preparation like chopping, peeling, mixing, etc. Whereas food skills include broader components such as food shopping, meal planning and budgeting. In the case of healthy cooking, there is not a standardized definition; therefore, many authors have defined it individually and imprecisely [29]. Raber et al. [26] proposed a healthy cooking model defined by the frequency of cooking that involves cooking at home and doing it from scratch, the use of healthy techniques such as avoiding high temperatures or the use of unhealthy fats, the minimum use of certain ingredients such as added sugars or processed foods, and the use of foods such as whole grains, olive oil, fruits and vegetables.

Further, the scientific literature in this field has pointed out the existence of various factors (economic, social, cultural, environmental and individual) that can serve as barriers or facilitators for the adoption of healthy eating habits [5,6,30,31,32,33].

Despite the direct influence of cooking on the development of these habits, relatively few studies have been conducted in the Spanish context that provide data regarding the perceptions and practices of young university students with respect to cooking and healthy eating. Most of these studies have been carried out outside of Spain, and in universities with organizations, student profiles and gastronomic traditions that are very different from those of Spain (e.g., the United States). However, scientific evidence suggests that the university period is the time during which many eating habits are consolidated that will last into adulthood [5]. Hence, it is essential to understand the context-specific influences on individual eating behaviors at this point in the life cycle in order to design effective and sustainable health promotion interventions [5,34,35].

On the basis of the above considerations, there is clearly a need to conduct qualitative research with the aim of exploring the perceptions held by young university students with regard to healthy cooking, along with the barriers that prevent this population from eating a healthy diet.

## 2. Materials and Methods

### 2.1. Study Design and Ethical Approval

A qualitative methodological approach was adopted in this study. The use of this type of methodology has proven useful for studying the factors and habits that affect health, as well as for evaluating and planning health services and policies [36,37]. This study was conducted following the ethical principles of the Declaration of Helsinki and Belmont Report. Before conducting the focus-group interviews, written informed consent was obtained from each participant.

### 2.2. Study Settings

This research was carried out at the University of Huelva (Spain) during the 2018–2020 academic year. The University of Huelva is a small-sized university (around 11,300 students) with particularly interesting sociodemographic characteristics. For instance, on the one hand, the socioeconomic level of the students is, in general, medium low. A large proportion of the students’ fathers (28.5%) have a primary level of education and work mainly in the tertiary sector (hotels, restaurants and tourism services) or are inactive (25.2%), whilst the mothers have a secondary level of education (30.9%) and are mainly inactive or unemployed (53.1%) [38]. On the other hand, the “food environment” of the university offers the services of a university canteen for the entire campus with a menu from Monday to Friday at a cost of EUR 5.50. There are also a total of 39 vending machines distributed by the different faculties. The machines provide coffee, sweetened soft drinks, water, pastries and salty snacks. There is a large shopping center with fast food restaurants next to the campus.

### 2.3. Sample

The sample used was intentional. According to sampling typology by Teddlie and Yu [39], a homogeneous sampling was applied. The choice of this type of sample was justified on: (1) the homogeneity of the sociodemographic characteristics of the Huelva university population described above, and (2) the suitability of this type of sample when using the focus group technique [40].

To ensure the relevance of the selected subjects with the research objectives, the following inclusion criteria sampling were established: (1) to be registered as a student at the University of Huelva during the 2018–2019 academic year; (2) to be residing habitually outside the family home for the first time; (3) to cook daily or occasionally during the week; (4) to be between 19 and 30 years of age. Additionally, the following exclusion criteria sampling was: (1) to be an Erasmus or Socrates student; (2) to be studying for a second university degree; (3) not having a sufficient linguistic level in Spanish; (4) to be a first-year university student or; (5) having specific training in cooking.

### 2.4. Recruitment of Participants

Information related to the research study was published on the Moodle platform websites of the subjects of different university degrees, requesting the voluntary participation of the students. Interested students sent an email to the research team and the team selected participants based on the established inclusion criteria sampling along with certain socio-demographic variables such as gender, course, and degree subject (Table 1).

The total number of students participating in the study was determined by applying the sampling saturation criterion [41].

### 2.5. Data Collection

Data collection was carried out between January–May 2019. The technique for data collection was the focus group. Prior to carrying out the focus groups, a semi-structured discussion guide was designed according to the basic dimensions on which the research objectives were set: (1) perception and meaning of cooking and healthy cooking; (2) facilitators and barriers to healthy eating and cooking; (3) self-perception of cooking skills. The script was developed from a scientific literature review, including items similar to those used in other relevant research studies on the subject [19,31,32,42].

Finally, a semi-structured discussion guide consisting of a battery of 14 open-ended questions was created (Figure 1). We decided to use open, rather than closed, items to encourage reflection and discussion by the participants in relation to the basic dimensions of analysis of the study.

The design of the semi-structured discussion guide was piloted through a focus group to test the understanding of the language and the relevance of the questions according to the research objectives. The pilot test lasted 45 min. After piloting the semi-structured discussion guide, 4 focus groups were carried out. The focus groups included 6 to 7 young university students, and lasted an average of 90 min.

All focus groups were conducted by two expert researchers—a moderator and an observer for the focus group sessions. Before starting the discussion in the focus groups, the participants were asked for their express consent to proceed with the audio recording of the session. The focus group discussion sessions were held in different classrooms of the university, making use of a circular spatial arrangement around a single table to encourage communication among participants.

### 2.6. Analysis

The information generated in the focus groups’ discussions was recorded in digital audio format. All recorded information was transcribed verbatim by the same researcher who moderated the focus groups. A quality control analysis was conducted on the transcribed information, which consisted of the random selection of different experts for verification.

The transcribed information was analyzed using the content analysis method [43]. The use of this method is based on the need to explore this socio-educational reality without explaining it on the basis of any preconceived theory. Rather, it is used to develop and describe concepts and hypotheses based on the data and categories that emerge from the discourses [44].

The inductive process of data analysis was [40]: (1) reading and rereading the focus group transcripts; (2) labelling the information according to the units, categories and codes of analysis; (3) describing the manifest and latent content of the categories and dimensions; (4) carrying out a relational analysis of the codes and categories, and then with concepts and theories.

All of the information was registered using the qualitative software ATLAS.ti version 8 (Scientific Software Development GmbH, Berlin, Germany).

### 2.7. Credibility

Following Dahlgren, Emmelin and Winkvist [45], the strategies used to increase the credibility of the study were: (1) Prolonged engagement: many of the research team members were teachers at the University of Huelva; (2) Multidisciplinary approach: the analysis of the information was carried out by researchers from different scientific disciplines; (3) Peer-debriefing: the preliminary results of the study were presented to investigators from outside the research process.

## 3. Results

### 3.1. Perceptions and Meanings of Healthy Cooking

#### 3.1.1. Perception of What Is Meant by Cooking

In general, the discourse of the young university students reveals the coexistence of two different perceptions with regard to the meaning of cooking. The first perception, held by most of them, was based on the consideration that cooking consists of a process of preparing or processing food from scratch, that is, using raw or fresh food to produce a final result that is edible and more appetizing:

“It is the process by which a product becomes edible or more palatable”(M, 25 years)

“It’s making food so that it can be eaten”(W, 21 years)

“Usually you start with the fresh fruit and then you process it, right?”(M, 24 years)

“Eating something more elaborate that tastes better than something you buy”(W, 22 years)

In this regard, the students explained that cooking is a more complex process than simply heating something (in the microwave or in the frying pan), and that it involves the use of different ingredients, techniques and utensils:

“It is a process that involves cutting food, seasoning, using spices, cooking it all together, something beyond heating the food, that is”(W, 22 years)

“You have to use some kind of equipment, because if you don’t, it’s not cooking”(M, 20 years)

Within this general perception, for some young university students, cooking is a daily priority and is synonymous with “eating well”. However, they mentioned that this entailed investing more effort and time, without which the result was not the same. These two elements have a positive connotation, insofar as they allow one to enjoy a richer meal, made by oneself, and to be familiar with the process of preparing food, that is, to know what one is cooking:

“It means knowing what I eat by making it myself and not buying it elsewhere”(W, 22 years)

“For me it’s something entertaining, I like to eat well, so I don’t like a precooked dish, I prefer to invest more time and have it come out tastier and say ‘‘I made it and it came out tasty”(W, 21 years)

Another element highlighted by some students is that cooking requires the use of certain prior knowledge (such as specific vocabulary) and skills such as cutting or peeling, which, although not very complicated, are necessary and take more time:

“I don’t know if that’s what you mean, but maybe cooking is something that requires a little skill, a little knowledge that you have to put into practice; it’s nothing super complicated but it requires a little more time”(M, 23 years)

“Sure, but if the recipe says to season such a thing and you don’t know what it is to season, you have to see how that “skill” is demonstrated in order to carry it out…”(W, 26 years)

Finally, for certain students, cooking was perceived as a process that also involves experimenting with flavors and mixing foods that develop the senses, something that is more linked to enjoyment:

“I think the same, it’s an activity that can be quite entertaining if you like to experiment with flavors too, mix them and it’s also quite important to have time because without it you obviously won’t be able to do it well”(M, 20 years)

“Also, for the pleasure I get from eating, because you eat a variety of foods, you try new flavors and you feel good about it”(W, 21 years)

The second, minority perception of what cooking is was based on the general consideration that cooking might consist of a simple activity such as heating something to make it edible, as is the case with raw, cold or frozen foods, or, because they are cold dishes, it may not even involve the use of heat or specific equipment:

“For me, cooking can be anything. Cooking with the pan, in the end, is making something for you to eat. If it’s frozen you can’t eat it, even if it’s just put it in the microwave without oil or anything, or a precooked dish that you have to put in the microwave that otherwise you can’t eat it either”(W, 21 years)

“You don’t always need specific equipment or to heat something; it depends on the food or your desired objective as to whether you use one or the other”(W, 22 years)

#### 3.1.2. Perception of What Is Meant by Healthy Cooking

In general terms, three elements defined healthy cooking for most of the students who participated in the study: (1) a complicated process; (2) healthy ingredients; (3) use of healthy techniques.

##### Complexity of the Process

Consistent with the general perception of process-based cooking, the students indicated that healthy cooking is also a process that involves complexity. Thus, to the extent that healthy cooking is perceived to involve a more elaborate process, they mentioned that it required more time:

“I consider that healthy cooking has a very complicated process behind it. Putting any kind of food in the microwave and making it fast, I don’t consider that it is healthy cooking but there is also the process of peeling the food, cutting it, making it first with medium heat and then raising it more, spending more time and of course with food that I know is healthy, I also have that previous knowledge”(W, 21 years)

“Making a stew is more laborious, and requires more time than buying processed food, getting it home and heating it up in the microwave”(W, 21 years)

##### Healthy Ingredients

This category emerges from the discourses of the students and represents the element that distinguishes between “cooking” and “healthy cooking”. In this regard, a series of qualities appeared in the discourses, which define these “healthy ingredients” such as being fresh, natural and unprocessed food:

“So that the ingredients you use are healthy”(M, 25 years)

“Belonging to the Mediterranean diet”(M, 25 years)

“Cooking with vegetables, fresh fish, meat. The more organic and natural the healthier”(W, 21 years)

The students also indicated which types of ingredients are unhealthy, such as sugars, saturated fats, salt and refined oils, and which therefore should not be used for healthy cooking:

“No ultra-processed products, they contain only what is food”(W, 22 years)

“Always use olive oil too, without too much saturated fat—do not use sunflower oil, control the sugar you include in your meals, or salt too…”(W, 22 years)

Furthermore, for some participants, healthy cooking was not only about the quality of ingredients but also about quantity, i.e., the proportions of food used in cooking, as well as variety:

“Also, the quantities, eating a little bit of everything, as the Mediterranean diet says, in my opinion. A little bit of everything, nothing but vegetables”(W, 21 years)

“It is true that you can cook a steak very well and eat it every day, but really you are not eating healthily, because you are not varying your food, but, for example, cooking fruit or vegetables, which someone may like more or like less, would be eating healthily, although the preparation is not very complex”(M, 22 years)

Finally, some students also commented on the importance of having prior knowledge about what is a healthy ingredient and what is not, as well as knowing the origin of the food they are going to cook:

“(…) and of course, with food that I know is healthy, I also have that prior knowledge”(W, 21 years)

“Therefore, use fresh food that we know where it comes from and that it is not processed”(W, 21 years)

“When I’m cooking I have to be careful and check that the food is not contaminated with trace elements”(W, 23 years)

##### Using Healthy Cooking Techniques

A third element that, for most young university students, represented healthy cooking, was the techniques used for cooking. In this regard, they explained that the use of certain techniques for preparing food, such as grilling, steaming, or baking—as opposed to techniques such as frying food—promote healthier cooking:

“I think the main thing is the way you cook it and also the food you cook”(M, 22 years)

“It’s about grilling or steaming”(W, 21 years)

“Using the oven rather than frying”(W, 22 years)

#### 3.1.3. Students’ Reasons for Healthy Cooking

Overall, healthy cooking is an important aspect for most young university students, regardless of whether they like to do it or not. Only a few students did not attach importance to this aspect:

“It’s important to me what I cook for myself, to cook it well and healthily”(W, 21 years)

“It’s important to me, even if I don’t like it”(W, 26 years)

“It’s not important to me”(M, 22 years)

“It’s important, but on what level? Because sometimes it’s not important enough to stop and make yourself food that’s healthy”(M, 20 years)

For most students, health was the main reason for healthy cooking. In this regard, some commented on the health problems they suffer from and hence the fact that healthy cooking was, for them, a necessity:

“Basically, for health, for trying to be healthy”(M, 26 years)

“Eating healthily mainly for health”(W, 21 years)

“Because when I’m 15 years old, if I get fed up with cooking, I don’t notice it, but... later on it takes its toll”.

“I usually eat well, because my stomach does not digest heavy things or things that are not very light or healthy”(W, 23 years)

“I have had cholesterol since I was little (…), so I have to try to keep it down and also because I have anxiety and tend to binge”(W, 21 years)

### 3.2. Barriers to Healthy Eating

Although eating healthily was viewed as an important aspect of health and well-being for most of the students interviewed, they did not usually do so:

“I consider it very important to eat healthily. The problem is that even if I know what is right and I know what I should do, I don’t do it, or I don’t do it as I should. Even though sometimes we think, “well, this food would be better or healthier if I did it this way”, due to lack of time we often resort to other ways that are not the correct ones, even though we know which one is the best. Although I do consider it super important”(W, 21 years)

“It’s important, but I’m not going to be a hypocrite, what happens is that maybe it’s important for you to be healthy, to be comfortable with yourself, but there are times when I skip it as I please. Of course, it is important to eat healthily, it is part of your health, but when it comes to putting it into practice, it seems that I don’t give it the importance that it really has”(W, 21 years)

In this regard, they commented on the existence of a number of impediments (of various types) that stand in the way of implementing a healthy eating regime. The ones that most affected all the young university students interviewed were:

#### 3.2.1. Economics

Among the various economic barriers to healthy eating, they identified three types: (1) the financial situation of the student, (2) the price of the food, and (3) the lack of equipment. With respect to the first, they explained that the student status entails (given that they do not work) not having financial independence and therefore a budget to be able to buy food:

“The financial situation, which is the life of the university student, will always be present”(W, 20 years)

“And we, because we are students, we select the cheapest”(W, 23 years)

“Economics, and also knowing that we have a supermarket close by, and because it’s cheap, it is going to be less healthy for sure”(W, 21 years)

Second, they pointed out the price of food and eating. In this regard, the students perceived that certain healthy foods have a higher price. Thus, the cost of organic products or fresh foods such as fish, as opposed to less healthy foods, made it difficult to buy them.

“If you buy something fresh, it’s healthier but more expensive”(M, 20 years)

“That’s right, organic food is more expensive”(M, 20 years)

“The price. I like fish a lot, but depending on what I buy each week, it’s a bit expensive”(M, 22 years)

In addition, some young university students commented that amongst the products stocked in the vending machines that are located at the university, the healthiest options, such as nuts, are the least affordable:

“Ultra-processed or unhealthy products on top are cheaper”(W, 23 years)

“I especially notice the price when it comes to snacks. I usually eat many times a day, so it’s not the same if you have a pack of cookies (which has cost you a little and you get a lot) as if you have a bag of nuts, which has cost you much more money. You can’t be eating in the middle of the morning, and in the afternoon, because it costs more money”(W, 22 years)

Finally, a third economic barrier they pointed out was the lack of equipment (cooking appliances and utensils):

“It’s true that utensils are very important when it comes to cutting or chopping… Try cutting a tomato with a knife that doesn’t cut, you kill it”(M, 23 years)

“It’s also important to have all the things you need to use”(M, 25 years)

#### 3.2.2. Time

A further problem perceived by most young university students was the lack of time available to cook regularly. This aspect was one of the main barriers to healthy eating, as they explained that healthy cooking requires much more time for food preparation and processing than the preparation of processed or pre-cooked food:

“That you take longer”(M, 29 years)

“The time you have to spend on it, which is always more than a pizza that you put in the oven and in 15 min it’s ready”(W, 23 years)

“Time basically. By having more time, you can prepare the food better and when you have less time you just grab what you can get”(W, 26 years)

In this regard, some participants questioned whether lack of time was the real reason, or whether this “lack of time” is instead due to a lack of organization and planning for cooking:

“I think it’s mostly about organization and time and also not feeling like making something healthy, or stopping to think about what you’re going to eat to make it healthy”(M, 20 years)

“I think that more than lack of time is lack of planning, at least in my case, if you want time you will find it. Maybe you have to plan what you’re going to spend more or less, when you’re going to cook, when you’re going to go shopping…”(W, 22 years)

One aspect related to the lack of time for healthy eating that was mentioned by some of the young university students is the established schedule of classes at the university, since the morning or afternoon sessions prevent them from cooking at midday:

“I arrive home at 2:30 pm and whilst I begin cooking, I get ready and no, it’s already 4 pm. Tell me when I can sleep, when I can study and when I can do everything”(W, 20 years)

“More than anything it’s time, because, for example, if you have classes and then you have to come back after eating, you can’t make a healthy meal because it takes longer than if you fry something or grill something”(M, 25 years)

#### 3.2.3. Willingness

For some young university students, another barrier was the lack of desire or willingness to cook, and therefore eat healthily. They said that sometimes the willingness to eat healthily was hindered by the temptations of prepared food offered by the food industry, and this is the case with the products offered in the food vending machines in the university environment:

“And not eating healthily because of cravings. A lot of times here, I see a candy truffle in the machines, and I think “come on, I’ll buy it””(W, 21 years)

“Will power, instead of eating something I really like, today, say “I’m going to eat this which is healthier””(W, 21 years)

“It’s mostly being too lazy to cook, not that I don’t have time”(W, 23 years)

#### 3.2.4. Geographical Accessibility

The students commented that one aspect that made healthy eating difficult was the lack of geographical accessibility when it comes to buying healthy food. In this sense, they explained that where they live there was greater accessibility to large stores in the neighborhood than to local markets, which ultimately determined the type of products they bought for cooking:

“If we have big supermarkets nearby, we go to those instead of going to the small one which is surely more expensive but also healthier. It depends on whether it’s on the way, let’s go for the comfortable option”(W, 21 years)

“Here in the city it is difficult to get vegetables and products from the town, because, for example, in my area we have a big supermarket next door, and so why would we go to a market?”(W, 20 years)

Some students also mentioned the lack of accessibility that existed in the university environment when it comes to purchasing healthy products:

“When I’m studying for exams at the library in the evening and I feel like eating something and haven’t brought anything with me, the only healthy option is a mini bag of nuts and all the other baked goods, sweets, etc.”(W, 21 years)

#### 3.2.5. Culinary Knowledge and Skills

Another barrier pointed out–albeit by a minority of participants—was the lack of culinary knowledge and skills needed to cook healthily:

“I don’t have time to start cooking healthily, and I don’t know how to do it either. The basics are the only things I know how to do”(M, 22 years)

“And now less so, but in the first years of my career, I didn’t know how to do anything. You’re in a hurry, you cook it fast, you don’t like the way it tastes, and you throw it away, and in the end, you leave without eating”(M, 25 years)

This lack of knowledge and culinary skills appears to generate a feeling of monotony in some students, and therefore boredom, so much so that they end up eating the same thing every time:

“For me it’s also important because I think that knowing how to cook healthily allows you to avoid becoming bored with the food. And that’s what happens to many of us, who end up eating the same thing all the time, and so it’s better to look for a healthy alternative than one that is less healthy, but because we don’t know how to do it, that’s what happens to us”(M, 22 years)

However, for other students, this lack of knowledge produced feelings of insecurity and lack of confidence in cooking:

“I go with the mindset of “this is not going to work out, go and order a Chinese …’’ ‘I’ve never done this before. “Maybe I’ll try it later and say, “Look how well that worked out for me, and it’s the first time”(W, 23 years)

“I have confidence in my abilities, but I have the insecurity of never having done it, to see how it will turn out, and sometimes it turns out better and other times worse, it is really about practice(W, 21 years)

#### 3.2.6. Emotions

Some students felt that certain emotions could act as a barrier to eating healthily. They explained that usually when they felt stressed, or were in a negative mood, they would eat unhealthy foods or products:

“I usually eat more bad food when I’m taking exams, because when I’m nervous I want more”(W, 23 years)

“When I eat badly it’s always because of stress or lack of time. You eat things that are quicker to make and have them at hand, but usually because of stress”(W, 23 years)

#### 3.2.7. Eating with Others

On the other hand, some students pointed out that eating in company could act as a barrier rather than a facilitator when it comes to healthy eating. For some students, eating out with friends or family or going to friends’ homes means eating convenience food or ready meals, which is therefore less healthy:

“For example, one Thursday I go out to a club and I go to my friend’s house: “Well, let’s have a pizza”, and I end up eating it”(W, 21 years)

“Another problem is when you are invited to socialize at other houses or when you go with your friends or family, as they usually go to McDonald’s, at least in my case”(W, 21 years)

## 4. Discussion

To the best of our knowledge, this is the first study to explore the perceived barriers to healthy cooking and eating in university students in the context of a small Spanish university. Regarding the perceptions about cooking—and healthy cooking—of the students of the University of Huelva, our results are consistent with the findings of previous studies [19,26,32]. First, our findings reveal several perceptions among students with regard to what is meant by cooking. This lack of unanimity in this definition can also be observed in the works of other authors [17,19,46]. Some explain this variability and lack of consensus by the fact that, in comparison with previous decades, cooking is currently subject to continuous changes due to the incorporation of new technologies and gadgets as well as the wider availability of pre-cooked products [47,48] making the practice of cooking a more dynamic and flexible activity [17]. Therefore, in line with the findings reported in other studies, for students, cooking is defined primarily in terms of a process [15,19] as opposed to the minority definition of cooking as a simple activity, which can be limited to the act of heating pre-cooked foods [17,19]. This shows how this stage of transition regarding concepts and habits influences and makes more flexible the perceptions held by the younger population [49].

Second, it should be noted that for most students, cooking is not synonymous with healthy cooking. There are, therefore, differences in perceptions about what is meant by cooking in general, and what is meant by healthy cooking. In this respect, students generally define healthy cooking as a more elaborate process that requires special ingredients and the use of healthier cooking techniques in comparison with cooking in general. Furthermore, it implies a perception of healthy cooking that requires more effort, time and skills than cooking in general. This definition of healthy cooking is compatible, for the most part, with the model developed by Raber et al. [26], which offers a conceptual framework for the term that includes categories such as techniques, skills or ingredients to be avoided and included for healthy cooking.

In order to design interventions aimed at improving the eating habits of young people, an important aspect to consider is the perceptions that currently exist about healthy cooking [26]. In our case, university students offer a definition of healthy cooking that implies barriers to its practice because, for them, such a definition suggests that it is a challenging and time-consuming process. Therefore, it is necessary to work on these false beliefs about healthy cooking when designing interventions aimed at improving the eating habits of university students.

Third, despite the importance of healthy eating habits for the preservation of good health, most students say they encounter various difficulties in putting such habits into practice. Following the model by Deliens et al. [31], which describes the factors influencing the eating behavior of students, our findings support the notion of an interaction between individual and external factors highlighted in other studies [5,6,19,31,33,50]. In our case, the type of barriers perceived as most influential are individual (budget, planning, willpower, cooking skills and emotional state) and environmental (food costs, equipment, class schedules and geographical accessibility), as opposed to social (the influence of peers and family members) barriers (Figure 2).

Certain individual barriers identified by the students, such as an insufficient budget for the purchase of healthy foods, have also been pointed out by other authors [5,32,50]. However, this limited budget interacts with the perception that healthy foods are more expensive, and is an environmental barrier that is most likely, as Marquis [51] has pointed out, to encourage prioritization of the purchase of cheap and fast foods as opposed to healthy and less affordable products [52]. In relation to the latter, the greater accessibility, through vending machines, of ultra-processed and prepared products that are cheaper in the university environment interacts with a lack of individual willpower, making it difficult to make healthy food purchases. In this regard, a number of studies have shown that exposure to an obese environment increases the likelihood that people will consume unhealthy foods [53,54].

Likewise, the students identified university-related environmental barriers, such as schedules that make it difficult to reconcile cooking time with class attendance, a finding that has also been reported by other authors [6,31,32] and is a factor that is particularly relevant for our young university students who perceive that healthy cooking is time-consuming.

Therefore, the existence of individual and environmental barriers that make it difficult for university students to put healthy eating habits into practice warns us of the need to plan and design interventions that consider individual factors, such as empowerment in self-regulation or self-discipline and stress management skills. In addition, it is important to modify the context to facilitate healthy decision-making, increase healthy options, make these options available at more appealing prices, and discourage unhealthy choices, which could include regulating the content of vending machines at the university. Finally, and more specifically, there is a need to consider the particular context of the young university students of the University of Huelva where, contrary to what usually occurs in other countries and/or universities in which students usually eat in buffets and in the dining rooms of the university residences, these students usually live in shared flats with other students and rarely in student residences. This means that for these students there is a greater responsibility to prepare the food they eat and to make decisions about what they buy.

In order to promote healthy eating, a relevant model for future interventions could be the one proposed in the food literacy framework [55]. It implies a comprehensive approach that takes into account individual factors such as knowledge and food skills or self-efficacy, and socio-ecological factors such as access to learning, kitchen equipment and other social determinants of health.

At an international level, relatively little research has been conducted on healthy cooking as a practice in the university population. Hence, one of the limitations of our research is the poor comparability of the results obtained from this dimension of analysis with other research outcomes.

Another limitation of the study has been the difficulty of accessing the group of young university students with a high socioeconomic level, since these students generally decide to enroll at universities located in other cities.

Likewise, another limitation derived from the use of this type of methodology is that the discourses of the university students can be influenced by what is socially desirable in terms of healthy eating habits.

Given that this is an exploratory study, future research should also include quantitative measures to identify the factors that are most likely to determine healthy eating habits in students.

## 5. Conclusions

Our findings contribute towards gaining a deeper understanding of student perceptions with regard to what is meant by healthy cooking, along with the barriers they face when trying to follow a healthy diet. The results obtained demonstrate the importance of what it means for the young university students to cook healthily as it constitutes a barrier both to put into practice and adopt a healthy diet. The study findings also show the existence of a variety of barriers of different kinds, namely, individual, social, economic and environmental. Accordingly, a holistic perspective should be considered for interventions to achieve healthy, feasible and sustainable changes in young university students eating habits.

## Figures and Tables

**Figure 1 nutrients-12-02309-f001:**
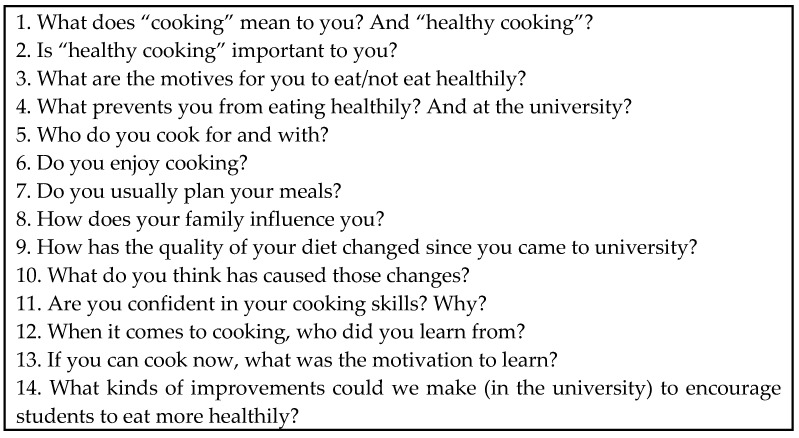
List of questions.

**Figure 2 nutrients-12-02309-f002:**
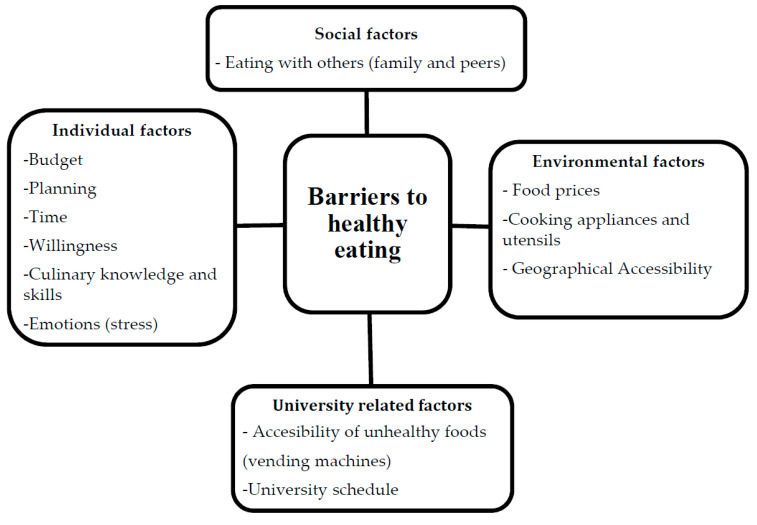
Perceived barriers to healthy eating among university students.

**Table 1 nutrients-12-02309-t001:** Socio-demographic characteristics of the participants (*n* = 26).

	Min–Max
**Age**	20–30
**Sex**	***n***
Female	14
Male	12
**Academic year of study**	
2nd	7
3rd	11
4th	8
**Degree**	***n***
Social Work	10
Agricultural Engineering	4
Forest Engineering	4
Psychology	4
Early Childhood Education	2
Nursery	2

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
