# Peer review of "“It’s Important but, on What Level?”: Healthy Cooking Meanings and Barriers to Healthy Eating among University Students"

_nutrients, 2020, doi:10.3390/nu12082309_

Round 1
Reviewer 1 Report
This is an interesting area of study and I agree with the authors it is one that is very much under researched. However, some work, in my opinion, is required before the article can be considered for publication.
The title of the article is not clear – “cooking perceptions” – a rewording of the title would enhance the article
Page 2 Line 50 – refers to decrease in time cooking but no up to date figures are given as a comparison to 2015.
Page 2 Line 51 there is a causal link stated between cooking hours and consumption of processed food in spain university students which is not substantiated
Page 2 Line 56 – there are definitions of cooking skills and healthy eating so not clear on what this statement is referring to that nothing exists.
A definition, from the literature, on what the researchers mean by “healthy cooking” needs to be outlined to contextualize the study. If this is the focus of the research then a starting point for a definition would need to be outlined. Additionally, some of the literature which differentiates cooking skills vs food skills could also be included.
The literature that is referenced, although relevant, could be enhanced be the addition of some more research published in recent years in this area.
Page 4 – there is huge disparity between the session times? 45 minutes to 120 minutes – what was the reason for this? Was the schedule piloted initially? Was it one person that conducted all focus groups?
The Findings section is clearly presented and relevant to the study. Acknowledging word limits, if there were space some more quotes could be included to substantiate the statements.
The discussion section of the paper could be further enhanced by including a more critical approach to the research.
Author Response
Dear reviewer:
We greatly appreciate your comments as they have favored the quality of the manuscript and have allowed us to broaden our view in certain aspects. We attach the cover letter with the changes.

Reviewer 2 Report
My major comments for the authors are as follows:
- In abstract:
- “ Positive correlation”- A very quantitative and statistically specific word. Please replace it.
- “ the time currently spent”- What does “currently “ mean here?
- “Semi-structured focus groups”- here semi-structure is a misleading word.
- Authors did not state the definition of healthy diet. This word has a variety of meanings and definitions.
- Internal validity is a term inclined towards positivistic paradigm, and therefore, more suited to quantitative studies. The equivalent term for qualitative genre is "credibility". Credibility refers to the extent to which a qualitative study could capture and reconstruct subjective realities of the participants.
- Please read the chapter titled "Designing qualitative research" in "Qualitative methodology for international public health" by Dahlgren, Emmelin and Winkvist (2007). Strategies for increasing credibility include: prolonged engagement with the setting/participants, triangulation, negative case analysis, peer-debriefing, member checks etcetera. Please revise the strategies written in Materials and Methods, as these are strategies for increasing internal validity in quantitative research.
- Please clarify: how to control quality of data analysis? Authors mentioned in line 156 “triangulated by different members of the research team”. However, triangulation consists of many methods. Authors referred a Spanish article (37), which is difficult to read by international readers. Please give the details. I am not sure whether authors consider about development of a coding system and inter-rater reliability or sending the transcripts back to participants to check the accuracy of meaning.
- Authors did not mention how they carried out the "inductive analysis", they did not refer to any article either.
- Authors described study setting in the topic of number of students in university and employment type of participants’ parent in line 82 -89. However, it will be better if authors can add the information about food in university. Food environment such as food stall or availability of food in university or community is an important factor for an individual’s eating habits.
- Line 557-558: This is actually not a limitation when it comes to qualitative methodology. In qualitative research the term equivalent to "generalizability" is "transferrability". Of note, qualitative researchers don´t intend to achieve "statistical generalizability", rather they intend to obtain "analytical generalizability". According to another school of thought, in qualitative research, transferability claims can never be made by the researchers, but should be made by the readers. Please see Dahlgren, Emmelin and Winkvist (mentioned above).
Author Response
Dear reviewer: We greatly appreciate your comments as they have favored the quality of the manuscript and have allowed us to broaden our view in certain aspects. We attach the cover letter with the changes.
Kind regards,
The authors

Reviewer 3 Report
This paper presents some new data about University students ' attitudes towards cooking and healthy eating. While it is not presenting particularly new results it contributes to this field of knowledge.
The main limits I have identified is the lack of characterisation of 'healthy' eating and practices of cooking. The paper could be improved by adding a clear description of how the authors defined healthy eating, whether or not they included culturally specific diets and foods and whether they gained an idea of the actual practices around food of the students they interviewed.
There is also a lack of awareness of some relevant literature on critical nutrition and race/ class. These are some relevant references:
Hayes-Conroy, A. (Ed.). (2016). Doing nutrition differently: critical approaches to diet and dietary intervention. Routledge.
Guthman, J. (2014). Introducing critical nutrition: a special issue on dietary advice and its discontents. Gastronomica: The Journal of Food and Culture, 14(3), 1-4.
Slocum, R., & Saldanha, A. (Eds.). (2016). Geographies of race and food: Fields, bodies, markets. Routledge.
Author Response

(The authors gave the same response as above.)

Round 2
Reviewer 1 Report
Thank you for making the suggested changes
Author Response
Dear reviewer:
Thank you so much for all your suggestions.
Best wishes,
Mercedes Vélez.
Reviewer 2 Report
I don't feel qualified to judge about the English language style. However, some revision by a native English speaker would be useful.
Author Response
Dear reviewer:
Thank you so much for your suggestions, specially for the methodological ones.
With respect to the English language, the manuscript has been proofread by a professional native speaker. However, we welcome any specific instance you have noted to improve the text.
Best wishes,
Mercedes Vélez.
Reviewer 3 Report
The authors have addressed some of my comments and have clarified that, given the nature of the sample of students that they recruited for the 4 focus group discussions (a very homogeneous group of students), issues of race, ethnicity, social class etc...are not examined and discussed.This is an inbuilt limitation of the study whose results cannot be generalised but should be considered an exploratory study of a very specific group of participants.
Not withstanding these limits the paper is improved.
Author Response
Dear reviewer:
Thank you so mucho for all your suggestions and your comments.
Best wishes,
Mercedes Vélez.